# Functional and structural dissection of glycosyltransferases underlying the glycodiversity of wolfberry-derived bioactive ingredients lycibarbarspermidines

Shao-Yang Li[1,2,4], Gao-Qian Wang[1,4], Liang Long[1,4], Jia-Ling Gao[1], Zheng-Qun Zhou[1], Yong-Heng Wang[1], Jian-Ming Lv[1], Guo-Dong Chen[1], Dan Hu[1] ✉, Ikuro Abe[3] ✉ & Hao Gao[1] ✉

Lycibarbarspermidines are unusual phenolamide glycosides characterized by a dicaffeoylspermidine core with multiple glycosyl substitutions, and serve as a major class of bioactive ingredients in the wolfberry. So far, little is known about the enzymatic basis of the glycosylation of phenolamides including dicaffeoylspermidine. Here, we identify five lycibarbarspermidine glycosyltransferases, LbUGT1-5, which are the first phenolamide-type glycosyltransferases and catalyze regioselective glycosylation of dicaffeoylspermidines to form structurally diverse lycibarbarspermidines in wolfberry. Notably, LbUGT3 acts as a distinctive enzyme that catalyzes a tandem sugar transfer to the ortho-dihydroxy group on the caffeoyl moiety to form the unusual ortho-diglucosylated product, while LbUGT1 accurately discriminates caffeoyl and dihydrocaffeoyl groups to catalyze a site-selective sugar transfer. Crystal structure analysis of the complexes of LbUGT1 and LbUGT3 with UDP, combined with molecular dynamics simulations, revealed the structural basis of the difference in glycosylation selectivity between LbUGT1 and LbUGT3. Site-directed mutagenesis illuminates a conserved tyrosine residue (Y389 in LbUGT1 and Y390 in LbUGT3) in PSPG box that plays a crucial role in regulating the regioselectivity of LbUGT1 and LbUGT3. Our study thus sheds light on the enzymatic underpinnings of the chemical diversity of lycibarbarspermidines in wolfberry, and expands the repertoire of glycosyltransferases in nature.

Wolfberry, also known as Goji berry, has been used as a well-known traditional Chinese medicine in China and other Asian countries for more than 2000 years[1]. It is also increasingly utilized as a famous edible species around the world due to its health-enhancing and anti-aging properties[2]. Wolfberry contains significant amounts of polysaccharides[3–5], carotenoids[6–8], polyphenols[9–11] and dicaffeoylspermidines[12–14], which are the major classes of bioactive ingredients in the wolfberry. Among them, dicaffeoylspermidine-based compounds featured by a spermidine core flanked by two caffeoyl-like moieties have been found in a variety of plants such as *Arabidopsis*[15], *Solanum quitoense* Lam[16], and *Scopolia tangutica*[17], and have a wide range of biological functions[18,19]. However, their

glycosylated forms were not identified until we recently revealed a group of dicaffeoylspermidine glycosides (named lycibarbarspermidines) as the major constituents in wolfberry (Supplementary Fig. 1), accounting for more than 0.2% of the dry weight and being suggested to possess potent anti-Alzheimer's disease (AD) and anti-oxidation activities[12,13].

Glycosylation, one of the most frequent chemical modifications in natural products, can significantly attenuate aglycone-related toxicity while enhancing hydrophilicity and pharmacokinetic parameters, thereby increasing the bioavailability of active molecules[20,21]. Glycosylation of the anti-cancer drug etoposide significantly reduces its toxicity[22], while glycosylation of puerarin results in a 100-fold increase in water solubility and enhanced anti-osteoporosis activity[23]. Moreover, glycosylation can directly contribute to the biological activity of natural products[20,21]. Mono-glycosylation of digitoxigenin results in a 6-fold increase in the binding affinity of sodium/potassium-ATPase[24], and glucuronosylation of glycyrrhetinic acid leads to the formation of a clinically useful medication, glycyrrhizin, for the treatment of chronic hepatitis[25]. As the main constituents of wolfberry, dicaffeoylspermidine glycosides are likely to retain vast potential for diverse biological functions.

Glycosylation in plants is generally mediated by uridine diphosphate (UDP)-dependent glycosyltransferases (UGTs), which catalyze the stereo- and regioselective transfer of an activated UDP-sugar to aglycones to form structurally diverse glycosides[26]. Up to date, numerous UGTs have been identified from plants, which catalyze the glycosylation of various molecules including terpenoids (UGT85C2)[27], flavonoids (F7GAT)[28], steroids (UGT74AN1)[29] and alkaloids (Betanidin 5-GT)[30] (Fig. 1A). However, the plant UGTs responsible for the glycosylation of phenolamides, which arise from phenolic moieties that are covalently linked to aromatic monoamine or aliphatic polyamine via amide bonds, such as dicaffeoylspermidines, have not yet been identified[19]. The discovery and elucidation of the UGTs underlying the glycodiversity of lycibarbarspermidine is therefore important to bridge this significant inventory and knowledge gap.

The lycibarbarspermidines identified from wolfberry exhibit diverse glycosylation patterns, particularly various di-glycosylation modes including 3′,3″-di-glycosylation, 3′,4′-di-glycosylation, 3′,4″-di-glycosylation, and 4′,4″-di-glycosylation (Fig. 1B, Supplementary Fig. 1)[12–14]. Despite the plethora of glycosylation profiles, lycibarbarspermidines consist of only four aglycones: $N^1$-caffeoyl-$N^{10}$-dihydrocaffeoylspermidine, $N^1$, $N^{10}$-bis-caffeoyl-spermidine, $N^1$-dihydrocaffeoyl-$N^{10}$-caffeoylspermidine and $N^1$, $N^{10}$-bis-dihydrocaffeoyl-spermidine[12]. Accordingly, the diverse glycosylation observed in lycibarbarspermidines could be influenced by the asymmetric nature of the middle spermidine core, the varying saturation degree of the caffeoyl moiety, and variations in the position of phenolic hydroxyl group. It is therefore interesting to investigate how these factors affect the glycosylation selectivity of lycibarbarspermidine UGTs, and explore whether these diglycosides are formed by a coordination of two specific UGT enzymes or by a single promiscuous UGT protein. Especially, an unusual di-glycosylation of the ortho-dihydroxyl group on the caffeoyl moiety, which is chemically difficult to accomplish due to steric hindrance, occurs in lycibarbarspermidines, suggesting that there exists an unprecedented UGT enzyme encoded in the genome of wolfberry.

In this study, we identify five lycibarbarspermidine glycosyltransferases (LbUGT1-5) from *Lycium barbarum*, which are the first phenolamide-type glycosyltransferases with distinct regioselectivity. Crystal structural analysis combined with molecular dynamics simulations and site-directed mutagenesis have illuminated the structural basis for the regioselective activities of LbUGTs. Our study thus uncovers the enzymatic basis of the glycodiversity of lycibarbarspermidines, and expands the enzymatic tools for glycosylation.

## Results

### Molecular basis underlying the glycodiversity of lycibarbarspermidines

The chemical diversity of lycibarbarspermidines is largely attributed to the diverse glycosylation of dicaffeoylspermidine aglycones. To identify the glycosyltransferases responsible for the glycosylation in lycibarbarspermidines, we systematically analyzed the UGTs within wolfberry. Since UGT71C1[31], UGT72E2[32] and TOGT1[33] were reported to glycosylate caffeic acid analogs, we used them as query sequences to screen potential UGT genes from the transcriptome of wolfberry. As a result, 151 LbUGTs were screened out and subjected to a phylogenetic analysis with known UGTs from plants (Supplementary Fig. 2). Accordingly, 20 LbUGTs were found to be close to the clade of caffeic acid glycosyltransferases. The candidate LbUGTs were cloned from *L. barbarum* cDNA into the pGEX-2TK vector and expressed in the *Escherichia coli* Rosetta (DE3) strain for functional characterization through in vitro enzymatic assays.

Since $N^1$-caffeoyl-$N^{10}$-dihydrocaffeoylspermidine (**1**) is the most abundant aglycone of lycibarbarspermidines, and contains both caffeoyl and dihydrocaffeoyl moieties, we first utilized **1** as the acceptor substrate and UDP-D-glucose (UDP-Glc) as the donor substrate by incubation with the crude extracts of the 20 LbUGTs expression strains for preliminary screening. As a result, five LbUGTs yielded products in the enzymatic reactions (Supplementary Fig. 3). Further liquid chromatography-mass spectrometry (LC-MS) analyses revealed that the newly generated compounds were 162 or 324 Da greater than that of **1**, indicating that the five LbUGTs could convert **1** into mono- or diglucosides, and thus were named LbUGT1-5.

To elucidate the catalytic properties and regioselectivities of LbUGT1-5, recombinant LbUGT1-5 proteins were purified and incubated with different aglycones. As shown in Fig. 2, LbUGT1 converts **1** to a major product **4** along with two minor products **5** and **6**, which were identified as monoglucosides LS-A (**4**), LS-B (**5**) and diglucoside LS-E (**6**), respectively, by comparison with authentic standards (Supplementary Fig. 4). These results indicated that LbUGT1 can catalyze the glycosylation at both the 4″-OH of a dihydrocaffeoyl moiety and the 3′-OH of a caffeoyl moiety. To clarify whether the regioselectivity of LbUGT1 is determined by the saturation state of the caffeoyl moiety or the asymmetric feature of the middle spermidine core (substitution at N1 or N10), we used two other acceptor substrates: N, $N^{10}$-bis-caffeoylspermidine (**2**) and $N^1$, $N^{10}$-bis-dihydrocaffeoylspermidine (**3**). The conversion of **2** into the major compound **9**, along with two minor compounds (**7**, **8**), was observed. These compounds were purified from large-scale reactions and identified as dicaffeoylspermidine 3′,3″-di-O-β-D-glucoside (**9**) and the monoglucosides **7** and **8**, respectively, according to a rigorous NMR analysis (Supplementary Tables 4–6)[34]. Similarly, **3** was converted by LbUGT1 to 4′,4″-di-O-β-D-glucoside LS-L (**12**) and its biosynthetic intermediates, 4′-mono-O-glucoside LS-H (**11**) and 4″-mono-O-glucoside LS-I (**10**), by comparison with authentic standards (Supplementary Fig. 4). These results clearly demonstrated that the regioselectivity of LbUGT1 is mainly affected by the saturation state of the caffeoyl moiety, and specifically catalyzes the glycosylation of 4-OH on a dihydrocaffeoyl moiety and 3-OH on a caffeoyl moiety, regardless of the (dihydro)caffeoyl moiety attached to N1 or N10 of the spermidine.

Intriguingly, both LbUGT2 and LbUGT4 catalyzed the production of a single monoglucoside, LS-D (**13**), at the 4′-OH of the caffeoyl moiety when using **1** as the substrate, although they share only 30% sequence identity (Fig. 3; Supplementary Fig. 4; Supplementary Table 2). Consistently, we observed that LbUGT2 and LbUGT4 converted **2** to the diglucoside **14**, but could not catalyze the transformation of **3** (Fig. 3). Although the products with an asterisk were not identified, they were presumed to be the monoglycoside intermediates of **14** based on the *m/z* value of 632 Da (Fig. 3). These results suggest that LbUGT2 and LbUGT4 specifically catalyze the glycosylation at the 4-OH of the caffeoyl group.

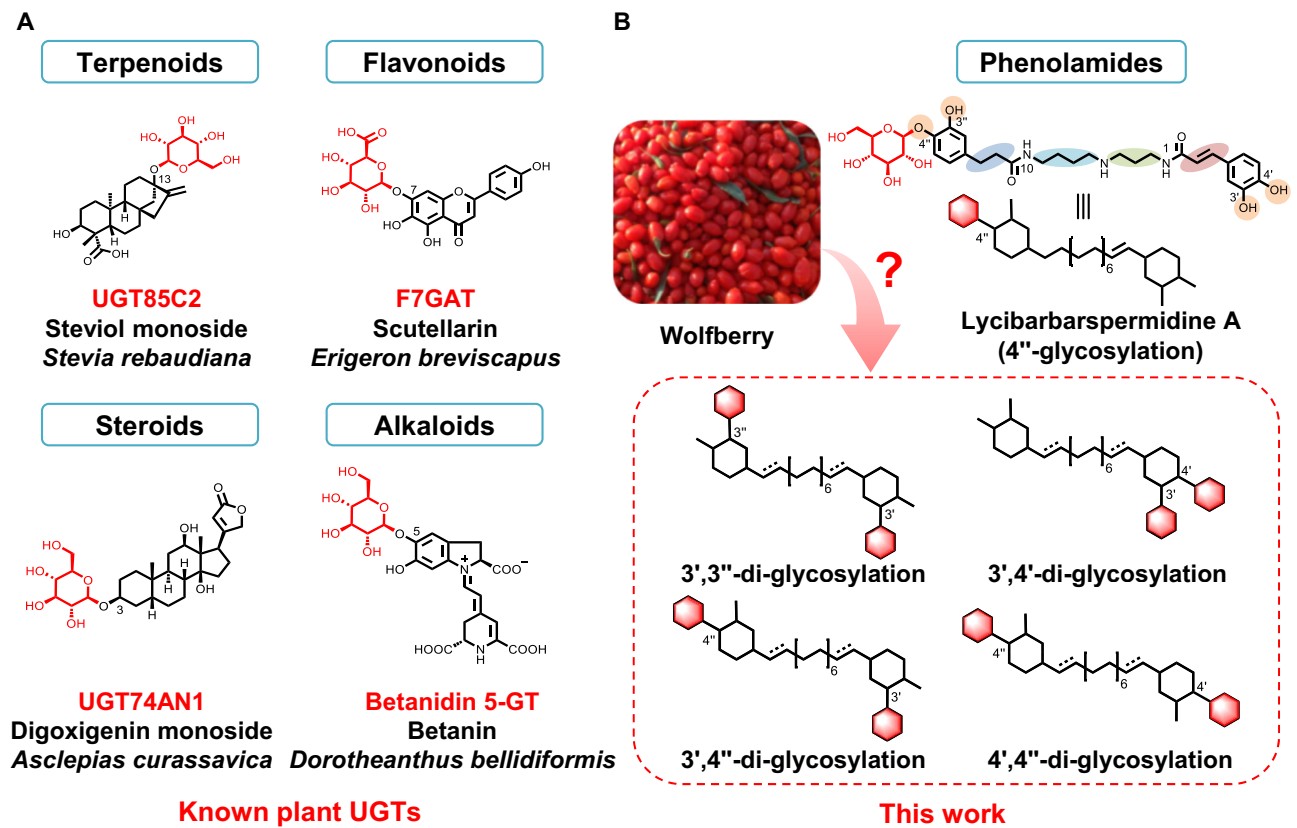

**Fig. 1 | The representative glycosyltransferases for diverse skeletal structures. A** Plant glycosyltransferases responsible for catalyzing terpenoid, flavonoid, steroid and alkaloid. **B** Representative phenolamide glycosylation product lycibarbarspermidines, and its disglycosylation modes cartoon representation.

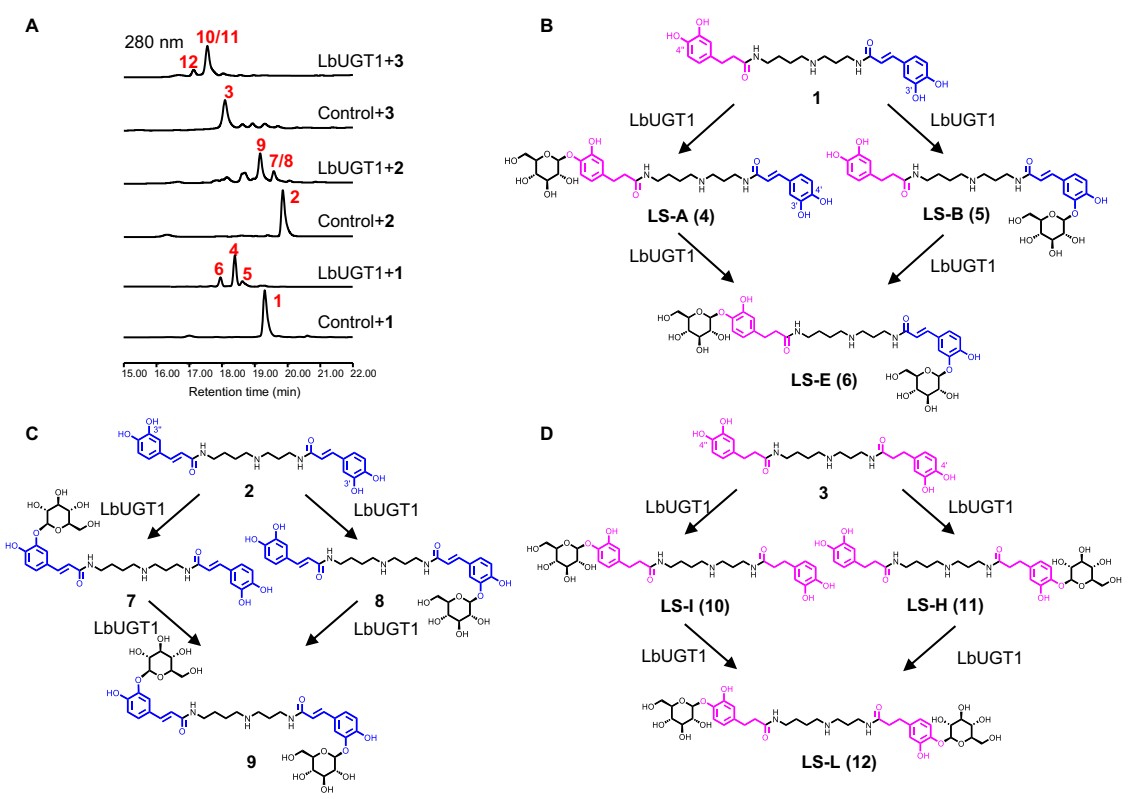

**Fig. 2 | Identification of LbUGT1 for the glycosylation of lycibarbarspermidines. A** HPLC profiles of LbUGT1 in vitro assays using **1, 2** and **3** as substrates. **B–D** Glycosylation reactions of LbUGT1 with **1, 2** and **3** as substrates, respectively.

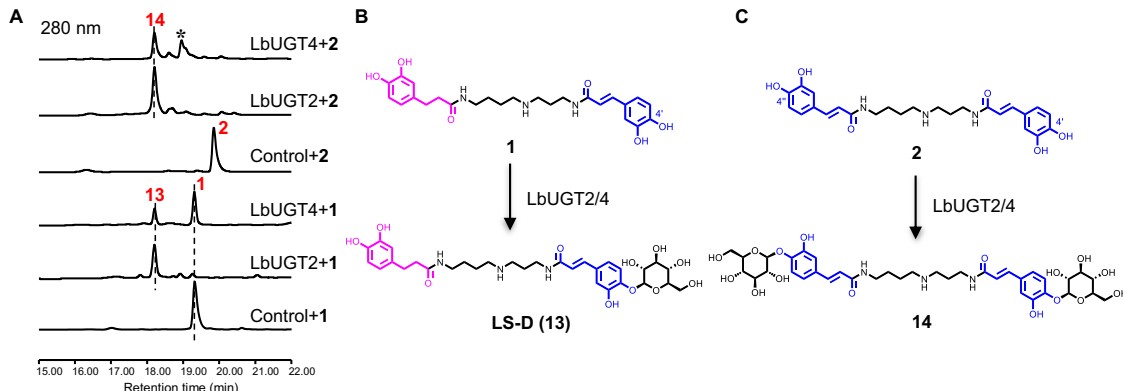

**Fig. 3 | Identification of LbUGT2/4 for the glycosylation of lycibarbarspermidines. A** HPLC profiles of LbUGT2/4 in vitro assays using **1** and **2** as substrates. The peaks marked with an asterisk are the presumed monoglycosides. **B**, **C** Glycosylation reactions of LbUGT2/4 with **1** and **2** as substrates, respectively.

LbUGT3 yielded the monoglucoside **5** and the unusual diglucoside **15** from **1** (Fig. 4). Compound **5** was identified as the 3′-*O*-glucoside LS-B (**5**) by comparison with an authentic standard, whereas **15** was isolated from scaled-up reactions and structurally determined to a novel diglucoside at 3′-OH and 4′-OH, according to the HMBC correlations of C-3′/H-1‴ ($\delta_C$ 147.2/$\delta_H$ 4.85) and C-4′/H-1‴′ ($\delta_C$ 148.4/$\delta_H$ 4.87), and the ROESY correlations of H-2′/H-1‴ ($\delta_H$ 7.38/$\delta_H$ 4.85) and H-5′/H-1‴′ ($\delta_H$ 7.16/$\delta_H$ 4.87) (Fig. 4; Supplementary Fig. 4; Supplementary Table 7). To verify the biosynthetic intermediate of **15**, enzymatic reactions of LbUGT3 were conducted using the 3′-*O*-glucoside LS-B (**5**) and 4′-*O*-glucoside LS-D (**13**) as substrates (Supplementary Fig. 5). The results showed that LbUGT3 converted both **5** and **13** into **15**, but preferred **5** over **13**, indicating that LbUGT3 mainly catalyzes the glycosylation at the 3′-OH to form **5**, and the 4′-*O*-glycosylation is its rate-limiting step for the di-glycosylation. In addition, LbUGT5, sharing ~40% sequence identity with LbUGT1 and LbUGT3 (Supplementary Table 2), also exhibited 3′-OH selectivity toward **1**, catalyzing the production of a small amount of LS-B (**5**) (Fig. 4). Like LbUGT1, LbUGT3 also converts **2** to monoglucosides **7** and **8**, and diglucoside 3′,3″-di-*O*-β-D-glucoside **9**, but it yields the unusual ortho-diglucosylated products dicaffeoylspermidine 3′,4′-di-*O*-β-D-glucoside (**17**) and 3″,4″-di-*O*-β-D-glucoside (**16**) as the major products (Fig. 4). Compounds **16** and **17** were prepared by large-scale enzymatic reactions and their structures were established by 2D NMR spectroscopic analyses (Supplementary Tables 8, 9). Though LbUGT5 also produce **7** and **8** from **2** like LbUGT3 (Fig. 4), neither LbUGT3 nor LbUGT5 catalyze the glycosylation of **3**. These results clearly indicate that LbUGT3 catalyzes an unusual tandem glycosylation of the ortho-dihydroxyl group of the caffeoyl group, while LbUGT5 specifically catalyzes the glycosylation of the 3-OH of the caffeoyl group. The enzyme LbUGT3, to the best of our knowledge, represents the first reported catalyst capable of facilitating the di-glycosylation on the ortho-dihydroxyl of a benzene ring.

The above results clearly indicated that LbUGT1-5 from *L. barbarum* display high regioselectivity toward the 3/4-OH of the caffeoyl or dihydrocaffeoyl groups, and are not affected by the asymmetry of the spermidine scaffold. The catalytic properties of LbUGT1-5 were further investigated. The recombinant LbUGT1-4 proteins showed their maximum activity at 30 °C when UDP-Glc as the sugar donor and **1** as the substrate, while LbUGT5 exhibited poor activity towards **1**, and substrate **2** was employed as the substrate and exhibited maximum catalytic efficiency at 37 °C (Supplementary Fig. 6). In addition, all five LbUGTs are divalent metal ion-dependent glycosyltransferases and exhibited maximum catalytic activity in the presence of Mg²⁺ (Supplementary Fig. 7). Due to the instability of the substrate in Tris-HCl and phosphate buffered saline, we could not achieve the optimal pH and all the experiments were performed in 10 mM HEPES buffer (pH

7.0). The enzymatic kinetic parameters of LbUGT1-5 were calculated based on Michaelis-Menten equations, and the $K_m$, $k_{cat}$, and $k_{cat}/K_m$ values are listed in Table 1 (Supplementary Fig. 8). The specificity of LbUGT1-5 towards different glycosyl donors was also investigated, as shown in Supplementary Fig. 9. LbUGT1/2/5 can utilize UDP-Gal similarly to UDP-Glc, while it does not accept UDP-GlcA, UDP-Xyl and UDP-GlcNAc as substrates. On the other hand, LbUGT3/4 specifically utilizes UDP-Glc as a glycosyl donor. These findings highlight the strong selectivity of LbUGTs for specific glycosyl donors.

Since LbUGT1 and LbUGT2 specifically catalyze the glycosylation of the 4-OH in the dihydrocaffeoyl and caffeoyl moieties, respectively, we speculated that the two enzymes should be responsible for the biosynthesis of $N^1$-caffeoyl-$N^{10}$-dihydrocaffeoylspermidine 4′,4″-di-*O*-β-D-glucoside lyciamarspermidine C (**18**), the most abundant lycibarbarspermidine within *L. barbarum*[12,14,35]. To explore this, we first incubated compound **4** with LbUGT2, and observed the transformation of **4** to **18** (Supplementary Figs. 4, 10). Similarly, the conversion of **13** to **18** by LbUGT1 was also observed (Supplementary Figs. 4, 10). These results suggest that the enzyme cascades of LbUGT1 and LbUGT2 could be used to generate the main lycibarbarspermidine **18**.

**Overall crystal structures of LbUGT1 and LbUGT3**
To explore the structural basis for the regioselectivities of LbUGTs, we tried to obtain the ternary complex structures of LbUGTs with UDP-Glc and different aglycone substrates. After a lot of trials including co-crystallization and soaking experiments, we obtained the complex crystal structures of LbUGT1/UDP at 2.57 Å resolutions (PDB ID: 8WP5; Fig. 5B) and LbUGT3/UDP at 2.43 Å resolutions (PDB ID: 8W53; Fig. 5C), respectively. The electron density for UDP in both structures was unambiguously assigned (Supplementary Fig. 11). LbUGT1 and LbUGT3 are the first phenolamide-type glycosyltransferases with crystal structures, and both exhibit a canonical GT-B fold consisting of two Rossmann-like fold (β/α/β) domains in the N and C-termini[36,37]. The N-terminal domain (residues 1−242 and 467−485 in LbUGT1, residues 1−246 and 464−482 in LbUGT3) is mainly involved in the binding of sugar acceptor, while the C-terminal domain (residues 243−466 in LbGT1, residues 247−463 in LbGT3) is responsible for the binding of sugar donor (Fig. 5B, C). Given that LbUGT1 and LbUGT3 share more than 50% amino acid sequence identity, it is not surprising that their structures are highly similar with a root mean square deviation (rmsd) of 1.04 Å over the 362 Cα atoms. However, an obvious difference can be observed in the N-terminal sugar acceptor binding region, especially in the putative substrate entry channel (residues 43−59, 68−86 in LbUGT1 and residues 45−62, 71−89 in LbUGT3) (Fig. 5D). This suggests that LbUGT1 and LbUGT3 may form different substrate entry/exit channel, enabling them to accommodate different substrates for sugar transfer. Structural alignment with other glycosyltransferases by the

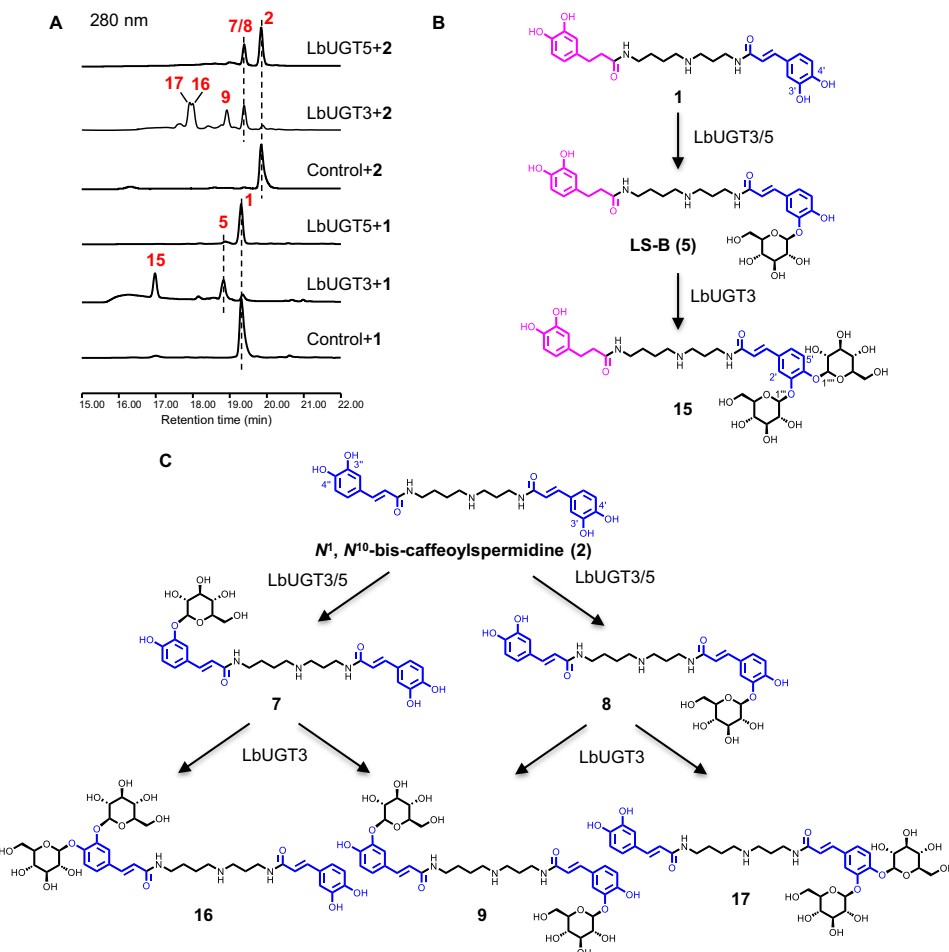

**Fig. 4 | Identification of LbUGT3/5 for the glycosylation of lycibarbarspermidines. A** HPLC profiles of LbUGT3/5 in vitro assays using **1** and **2** as substrates. **B**, **C** Glycosylation reactions of LbUGT3/5 with **1** and **2** as substrates, respectively.

Dali server shows that LbUGT1 and LbUGT3 share high structural similarity with the flavonoid/triterpene OGT UGT71G1 from *Medicago truncatula* (PDB ID: 2acw)[38] and the flavonoid OGT UGT74AC2 from *Siraitia grosvenorii* (PDB ID: 7bv3)[39]. They all harbor a catalytically important His residue (H17 in LbUGT1 and H19 in LbUGT3) in the active site, and a highly conserved C-terminal sugar donor binding domain, particularly the plant secondary product glycosyltransferase (PSPG) motif for UDP binding. These structures have allowed us to correctly model the glucose moiety of UDP-Glc into LbUGT1 and LbUGT3 for further mechanistic investigation (Supplementary Fig. 12).

**Structural basis of the regioselectivity of LbUGT1 and LbUGT3**
Though both LbUGT1 and LbUGT3 first catalyze the 3′-OH glycosylation of the caffeoyl group, LbUGT3 can further catalyze the 4′-OH glycosylation on the same side of **2** to form an unusual ortho-

diglucosylated product, while LbUGT1 needs to rotate the mono-glucosylated product for a sugar transfer to the 3″-OH on the opposite side (Fig. 5A). To elucidate the structural basis for the difference in regioselectivity of LbUGT1 and LbUGT3, the substrate **2** was docked into LbUGT1 and LbUGT3 based on the two key distances between the hydroxyl H atom of the acceptor substrate and the N atom of the catalytic histidine (d1), and between the hydroxyl O atom of the acceptor substrate and the C1 atom of UDP-Glc (d2) (Fig. 5A). By molecular dynamics (MD) simulations, we found the reactive binding conformations of **2** in LbUGT1 and LbUGT3 as verified by the small d1 and d2 values during the 50 ns simulations (Supplementary Fig. 13, Supplementary Data 1). In both models, H17 of LbUGT1 and H19 of LbUGT3 serve as essential catalytic residues[37,40], which were experimentally verified by the inactivity of the H17A and H19A variants (Fig. 5I). Interestingly, comparison of the binding modes of **2** in LbUGT1 and LbUGT3 revealed significant differences in the catalytic regions. The caffeoyl group of **2** in LbUGT3 sits in a hydrophilic cavity, wherein the 3′-OH side is encompassed by R321, S285 and C14, which is likely to facilitate the binding of the 3′-*O*-glucosyl group for further glycosylation on the adjacent 4′-OH. In contrast, LbUGT1 creates a hydrophobic residue-based pocket surrounding the 3′-OH side of the caffeoyl ring with bulky nonpolar F124, F86, L44 and P13, thereby obstructing the binding of the mono-glucosylated product and permitting only one glucose attachment to one end of the aglycone (Fig. 5E, F). Therefore, the presence of a hydrophilic pocket in LbUGT3 is essential for promoting excess sugar binding and enhancing their adaptability through hydrophilic

**Table 1 | Catalytic parameters for LbUGT1-5 using UDP-Glc as the sugar donor**

| | UGT names | Substrate | $K_m$·(μM) | $k_{cat}$·(s⁻¹) | $k_{cat}$·/$K_m$·(s⁻¹·M⁻¹) |
|---|---|---|---|---|---|
| LbUGT1 | UGT71AT6 | **1** | 259.4 | 5.7 ×10⁻³ | 22.1 |
| LbUGT2 | UGT73A52 | **1** | 761.6 | 13.4 ×10⁻³ | 17.6 |
| LbUGT3 | UGT71BG1 | **1** | 325.0 | 4.5 ×10⁻³ | 13.9 |
| LbUGT4 | UGT72B75 | **1** | 900.9 | 4.2 ×10⁻³ | 4.6 |
| LbUGT5 | UGT71AJ4 | **2** | 542.0 | 5.0 ×10⁻⁴ | 0.9 |

The source data underlying Table 1 are provided in a Source Data file.

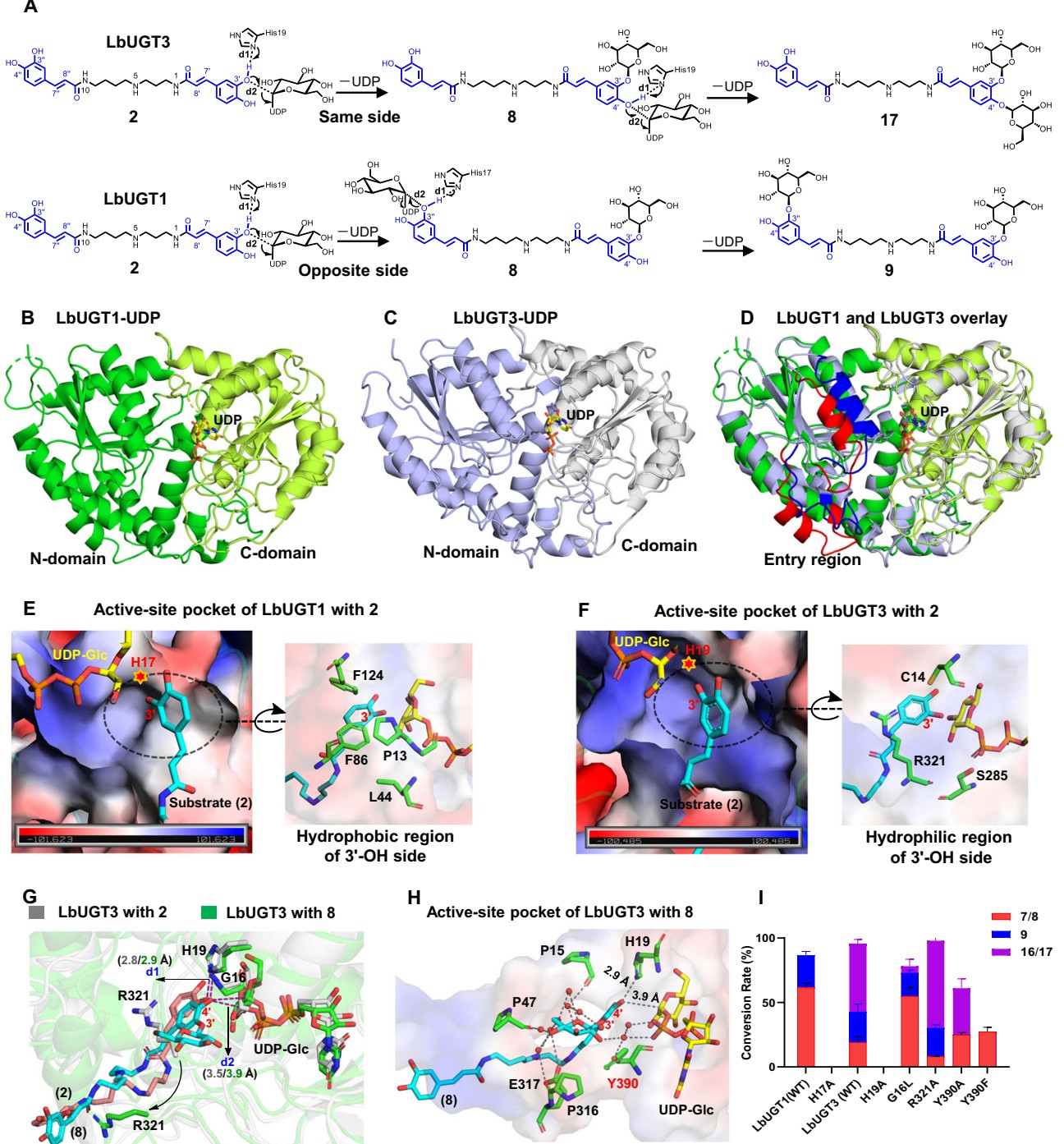

**Fig. 5 | Crystal structures of LbUGT1 and LbUGT3 and their regioselectivity analysis. A** Proposed catalytic mechanism for di-glycosylation of LbUGT1 and LbUGT3; **B, C** The crystal structures of LbUGT1/UDP and LbUGT3/UDP with the N-sugar acceptor binding domain (green and slate) and C-sugar donor binding domain (limon and gray). The UDP is shown as yellow sticks; **D** Overlay of LbUGT1/UDP and LbUGT3/UDP. Residues are color-coded as in **B. C** But the difference for the putative entry region colored in red and blue, respectively; **E, F** The active-site pockets of LbUGT1 and LbUGT3 with the common substrate **2** shown in cyan and UDP-Glc shown in yellow. The active-site pockets are shown in electrostatic surface representation (the negatively charged regions are shown in red, the positively charged regions are shown in blue and the hydrophobic regions are shown in white), the residues located within the circled region are colored as green sticks, residues His are represented by a red star; **G** Superposition of LbUGT3-**2** (residues and UDP-Glc, gray; **2**, salmon) with LbUGT3-**8** (residues and UDP-Glc, green; **8**, cyan). The hydrogen bonds were shown with the purple dash; **H** The active-site pocket of LbUGT3 with **8** (residues, green; UDP-Glc, yellow; **8**, cyan; hydrogen bonds, gray dash); **I** In vitro assay of the mutants of LbUGT1/3 using **2** as substrate. Data represent mean ± SD (*n* = 3). The source data underlying Fig. 5I are provided in a Source Data file.

interactions to overcome steric hindrance caused by adjacent di-glycosylation.

To further explain the regioselectivity mechanism of LbUGT3, the 3′-*O*-monoglucoside **8** was also docked into the protein structure and subjected to MD simulations to obtain the stable binding conformation (Supplementary Fig. 13, Supplementary Data 1). By comparing the two binding modes with aglycone **2** and monoglucoside **8** in LbUGT3, it is evident that the glucose moiety is just located in the hydrophilic

region. Notably, the flexible R321 exhibits a significant conformational alteration, potentially facilitating the expansion of sugar-binding pocket through adaptive shift (Fig. 5G). Furthermore, when the R321 was replaced by a smaller alanine, the yield of di-glycosylation products **16** and **17** increases dramatically (Fig. 5I), suggesting that modulation of the hydrophilic pocket size influence the ortho-diglycosylation function of LbUGT3. To confirm this hypothesis, we selected the G16 with the shortest side chain situated in the most prominent position on the hydrophilic pocket. After substituting G16 with the bulky nonpolar leucine, there was a significant reduction in the production of neighboring disaccharide products **16** and **17** (Fig. 5I). These results suggest that changing the size of the hydrophilic pocket exerts a substantial effect on ortho-diglycosylation of LbUGT3.

Also of note is that, the hydrophilic pocket surrounding glucose linked by **8** is filled with a cluster of ordered water molecules fixed by nearby backbone amides and carbonyl oxygens (like P15, P47, P316, E317). These water molecules form hydrogen bonds with 3′-O-glucose, and help bring the 4′-OH group closer to the catalytic residue for the second glycosylation (Fig. 5H). Additionally, a specific residue Y390 can engage in π-π interactions with the aglycone benzene ring and make water-mediated hydrogen bonds to the sugar moiety of **8** and phosphate group of UDP, likely play crucial role in determining the substrate binding conformation (Fig. 5H). Consistently, substitution of Y390 with F, which possesses a similar side chain to Y but lacks the phenolic hydroxyl group, completely eliminated the di-glycosylation activity despite of the decreased overall activity (Fig. 5I), highlighting the crucial role played by the hydroxyl group of Y390. Interestingly, substitution of Y390 with A still retained the di-glycosylation activity, which might be attributed to the enlarged channel size.

LbUGT1 is the only UGT that can recognize both the dihydrocaffeoyl and caffeoyl moieties, selectively catalyzing a sugar transfer to the 3′-OH of the caffeoyl group and the 4″-OH of the dihydrocaffeoyl group. This implies that the dihydrocaffeoyl moiety and the caffeoyl moiety might adopt different binding modes when entering the active site of LbUGT1 (Fig. 6A). In order to elucidate the molecular mechanism, the acceptor substrate **1** was docked into LbUGT1 structure, the MD simulations revealed the two active binding modes (Fig. 6B, C, Supplementary Data 1), corresponding to 4″-O-glycosylation (mode 1) and 3′-O-glycosylation (mode 2), respectively, which were verified by the small d1 and d2 values during the 50 ns simulations (Fig. 6D, E, left). Moreover, the average value of d2 for binding mode 1 (3.30 Å) is smaller than that for binding mode 2 (4.36 Å), consistent with the higher catalytic activity towards the 4″-O-glycosylation than the 3′-O-glycosylation by LbUGT1, which further confirms the reliability of our MD simulations. In addition, both binding modes showed several negatively charged residues, i.e., aspartic acids and glutamic acids (E83, D197, D416, E418), reside near the positively charged N5 of the substrate **1** (protonated in physiological environment). These residues may stabilize the conformation of the substrate through electrostatic or salt bridge interactions, and thus could play a crucial role in recognizing the spermidine unit of the substrate (Fig. 6B, C). Substitutions of these residues with A led to substantially decreased activity or even complete inactivation, confirming the above speculation (Fig. 6F). Furthermore, the inability of LbUGT1 to glycosylate caffeic acid or dihydrocaffeic acid may be attributed to its specific recognition of the spermidine unit in lycibarbarspermidines (Supplementary Fig. 14).

Comparisons of the two binding modes revealed that the residue interactions are essentially identical except for Y389, which forms a hydrogen bond with the 3″-OH of the acceptor substrate in binding mode 1, but not in binding mode 2, suggesting that Y389 might play an important role in stabilizing binding mode 1 for the 4″-OH glycosylation of the dihydrocaffeoyl moiety. Y389, similar to the corresponding important residue Y390 in LbUGT3, is located within the PSPG box, which is generally considered to be responsible for binding the sugar

donor[41–43]. However, recent reports showed that several residues of PSPG are critical for binding the acceptor molecules and altering the regioselectivity of UGTs[39,44,45]. To determine the potential role of Y389, Y389 and several PSPG residues that form hydrogen bonds to the UDP-sugar donor were mutated. Most of these mutants lost their activities, consistent with their important roles in stabilizing the sugar moiety and the glycosylation reaction. In contrast, the mutant Y389A remarkably changed its regioselectivity by eliminating its activity towards the 4″-OH but enhancing its activity at the 3′-OH. This is highly consistent with the two binding modes obtained by our MD simulations. Additionally, the mutation of Y389 to F also showed increased 3′-O glycosylation activity and impaired 4″-O glycosylation activity (Fig. 6F), indicating the important role of the OH group of Y389 in the 4″-O glycosylation activity.

To further confirm these results, MD simulations were also performed with the mutant Y389A for both binding modes and the two key distances, d1 and d2, were also measured. As shown in Fig. 6D, E, both d1 and d2 are stable within 50 ns for wild type LbUGT1 in modes 1 and 2 (Fig. 6D, E, left). However, the mutation of Y389 to A disrupted the hydrogen bond with the substrate in mode 1, and both d1 and d2 increased strikingly after 30 ns. In contrast, no effects of Y389A on mode 2 were observed (Fig. 6D, E, right). The above results further confirmed the importance of Y389 in the 4″-O glycosylation activity.

In contrast to LbUGT1 and LbUGT3, LbUGT2 and LbUGT4 catalyze the selective glycosylation of the 4-OH of the caffeoyl group. To investigate their structural basis, we used AlphaFold2 to model the structures of LbUGT2 and LbUGT4 due to the unavailability of their crystals. Following the methods used for LbUGT1 and LbUGT3, UDP-Glc and substrate **2** were docked into the protein structures, followed by MD simulations to acquire the stable binding conformations (Supplementary Fig. 15, Supplementary Data 1). Our findings show that in both LbUGT2 and LbUGT4 structures, the 4′-OH group of **2** forms a stable hydrogen bond with the catalytic histidine (H16 and H21) and is in close proximity to the acetal C atom of UDP-Glc, consistent with their glycosylation activity towards the 4-OH of the caffeoyl group. Notably, F382 in LbUGT2 and Y388 in LbUGT4, which correspond to the Y389 in LbUGT1 and Y390 in LbUGT3, engage in hydrophobic interaction and hydrogen bond with **2**, respectively. Subsequently, F382 in LbUGT2, Y388 in LbUGT4 and the corresponding residue Y384 in LbUGT5 were chosen for alanine-scanning mutagenesis. These mutations drastically decreased or even abolished their catalytic activities, indicating a crucial role for this conserved tyrosine residue in the function of LbUGT enzymes (Fig. 6F). Considering the high specificity of native LbUGT2, LbUGT4 to the 4-OH, and LbUGT5 to the 3-OH of the caffeoyl group, it is not surprising that these mutations do not give alternative products.

## Discussion

Lycibarbarspermidines are a major class of bioactive ingredients in wolfberry, whose chemical diversity is largely attributed to glycosylation. Here we identify the five glycosyltransferases (LbUGT1-5) underlying the molecular basis of the glycodiversity of lycibarbarspermidines. LbUGT1-5 from wolfberry are the first phenolamide-type glycosyltransferases, which catalyze regioselective glycosylation of dicaffeoylspermines to form structurally diverse lycibarbarspermidines. Although numerous plant UGTs have been reported to glycosylate a wide range of natural products such as polyphenols, terpenoids, alkaloids, UGTs that catalyze sugar transfer to phenolamides have not been reported[19,21,40]. The fact that LbUGT1 is unable to glycosylate free caffeic acid or dihydrocaffeic acid indicates that the interaction with the spermidine moiety is essential for the activity (Supplementary Fig. 14). Consistently, the modeled complex of LbUGT1 with aglycones revealed that the spermidine unit forms extensive interactions with several negatively charged residues (E83, D197, D416, E418). Mutation of these residues significantly impaired or eliminated the activity (Fig. 6F). On the other hand, LbUGTs do

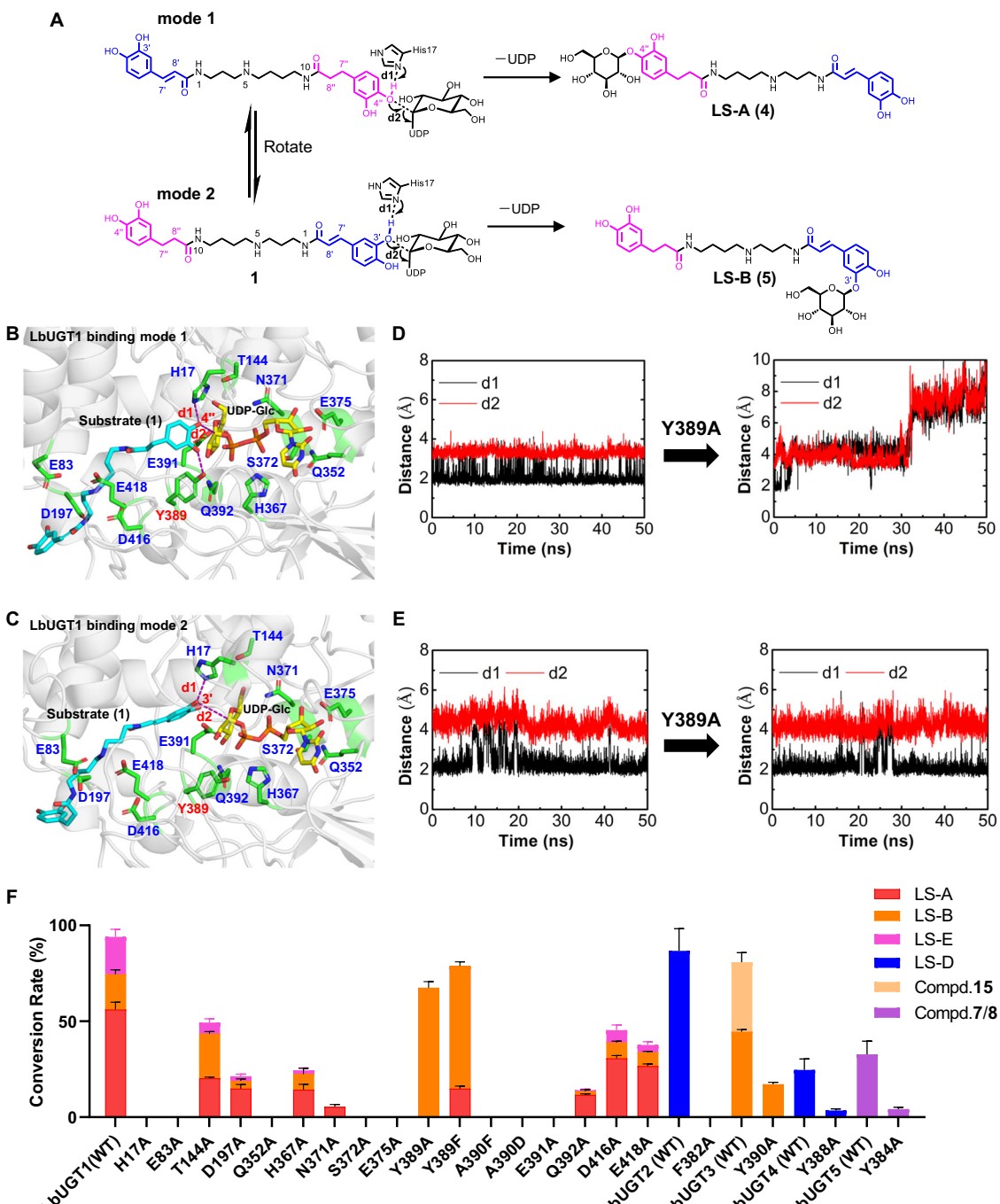

**Fig. 6 | Investigation of the catalytic specificity and the mechanism for LbUGT1.**
**A** Proposed catalytic mechanism for the 4″/3′-*O*-glycosylation of LbUGT1; **B**, **C** The binding mode 1 and binding mode 2 of LbUGT1 with the ligand **1** shown in cyan and UDP-Glc shown in yellow, the hydrogen bond was shown with the purple dash; **D**, **E** Time evolutions of the two key distances between the hydroxyl H atom of substrate and the N atom of the catalytic histidine (d1) and between the hydroxyl O atom of substrate and the acetal C atom of UDP-Glc (d2) in the 50 ns MD simulations of wild type LbUGT1 and mutant Y389A in two binding modes shown in **B** and **C**, respectively; **F** Exploring the catalytic specificity of variants of LbUGT1-5. Data represent mean ± SD (*n* = 3). The source data underlying Fig. 6F are provided in a Source Data file.

not accurately discriminate the asymmetric nature of spermidine, catalyzing glycosylation of caffeoyl groups attached to either N1 or N10 of the spermidine. The extensive adaptation of LbUGTs to the aliphatic polyamine portions may be utilized to glycosylate other phenoamides, such as kukoamine B, a promising therapeutic agent for type II diabetes mellitus treatment[46]. As expected, the conversion of kukoamine B is efficiently catalyzed by LbUGT1, resulting in the formation of monoglycosides with an increase of 162 Da (Supplementary Fig. 16).

The regioselectivity of LbUGT1-5 is significantly influenced by the saturation status of the caffeoyl groups. LbUGT1 selectively glycosylates the 3-OH of caffeoyl group and the 4-OH of dihydrocaffeoyl group, while LbUGT2-5 exclusively glycosylate caffeoyl groups. The strict regulation by the caffeoyl saturation is relatively uncommon, as most glycosyltransferases are influenced by the substrate backbone and the spatial position of acceptor groups. For example, three glycosyltransferases derived from *Epimedium koreanum* catalyze the 3-OH glycosylation of both kaempferol and dihydrokaempferol[47].

CsUGT75L12 from *Camellia sinensis* catalyzes the 7-OH glycosylation of apigenin as well as naringenin[48]. Our findings indicate that even subtle variations in saturation status of caffeoyl moieties can effectively modulate the recognition pattern and regioselectivity of glycosyltransferases, highlighting the remarkable precision and ingenuity exhibited by LbUGTs.

The enzymes LbUGT1-4 possess the ability to catalyze continuous di-glycosylation of dicaffeolyspermidines. Successive di-glycosylation has been frequently observed in multifunctional glycosyltransferases (Supplementary Fig. 17). For instance, UGTPg101 continuously catalyzes the glycosylation of protopanaxatriol at 6-OH and 20-OH to form protopanaxatriol Rg1, in which the two glycosylation sites are far apart from each other[49]. A similar strategy is also employed by UGT74F1, which simultaneously glycosylates 7-OH and 4′-OH of quercetin[50]. On the other hand, AmGT8 functions as a glucose chain extender that continuously catalyzes the glycosylation at 3-OH of cycloastragenol and 2′-OH of cycloastragenol-3-*O*-glycoside to produce astragaloside III[44], and CaUGT3 facilitates continuous glycosylation of the 3-OH and 6′-OH groups of quercetin 3-*O*-gentiobioside[51]. However, unlike the above cases, LbUGT3 is capable of catalyzing the di-glycosylation of the ortho-hydroxyl groups on the benzene ring, which exhibits higher steric hindrance due to the coplanar orientation of the phenolic groups. Accordingly natural products with ortho-glycosylation on a benzene ring are exceedingly rare, with only one case reported thus far[52].

Analysis of the active pockets of LbUGT1 and LbUGT3 identified the conserved residue Y389 (Y390) critical for the regiospecificity. Y389 in LbUGT1 or Y390 in LbUGT3 is the fourth residue from the C-terminus of the PSPG box (Supplementary Fig. 18). The third residue was recently reported to be crucial for the regioselectivity in AmGT8, a glycosyltransferase responsible for triterpene glycosylation in the medicinal plant *Astragalus membranaceus*[44]. We thus constructed the A390F and A390D mutants of LbUGT1 corresponding to those of AmGT8. However, the resulting mutants both lost the original activity and did not affect the regioselectivity (Fig. 6F), suggesting that LbUGT1 and LbUGT3 adopt distinct mechanisms for substrate recognition. The site Y389 or Y390 is highly conserved in other LbUGTs, in addition to LbUGT1 and LbUGT3 (Supplementary Fig. 18). The inactivity upon mutation in these glycosyltransferases suggests the essential role of the conserved tyrosine residue in glycosylation reaction (Fig. 6F). In addition, Y389 or Y390 is highly conserved in other plant glycosyltransferases (Supplementary Fig. 18)[31-33,44,53-55], suggesting an important role of this residue in UGTs other than LbUGTs.

In summary, we have functionally identified and structurally characterized the five phenolamide-type glycosyltransferases (LbUGT1-5), which account for the glycodiversity of lycibarbarspermidines that serve as a major class of bioactive ingredients in wolfberry. Among them, LbUGT3 can overcome strong steric hindrance to catalyze an unusual ortho-diglucosylation on the caffeoyl benzene ring, while LbUGT1 accurately discriminates caffeoyl and dihydrocaffeoyl to catalyze a site-selective sugar transfer. Comparison of the crystal structures of LbUGT3 with LbUGT1 reveals that a more hydrophilic catalytic pocket in LbUGT3 plays a pivotal role in facilitating ortho-diglycosylation. Site-directed mutagenesis identified a single conserved tyrosine residue is crucial in discrimination between the double bond and the single bond by LbUGT1. Our study shed light on the molecular basis for regioselective glycosylation of lycibarbarspermidines in wolfberry, and expands the enzymatic tools for glycosylation of biologically important natural products.

## Methods

### General materials and experimental procedures

Methanol (MeOH) was purchased from Yuwang Industrial Co., Ltd. (Yucheng, China). Analytical grade ethyl acetate (EtOAc) was from Fine Chemical Co., Ltd. (Tianjin, China). Formic acid and trifluoroacetic acid were obtained from Kemiou Chemical Reagent Co., Ltd. (Tianjin, China). Primer synthesis and DNA sequencing were performed by Sangon Biotech Co., Ltd. (Shanghai, China). Plasmid extraction kits and DNA purification kits were purchased from Sangon Biotech Co., Ltd. (Shanghai, China). PCR was performed using KOD One™ PCR master Mix (Toyobo, Japan). A ClonExpress® Multis One Step Cloning Kit (Vazyme, China) and an In-Fusion® HD Cloning Kit (Takara, Japan) were used to construct recombinant plasmids. DNA restriction enzymes and other DNA modification reagents were purchased from Thermo Fisher Scientific (Shenzhen, China). Caffeic acid, dihydrocaffeic acid and kukoamine B were purchased from Yuanye Bio-Technology Co., Ltd. (Shanghai, China). Compounds **1**–**6, 10**–**14** and **18** were from our laboratory's compound library.

HPLC-MS analyses were performed using a Dionex UltiMate 3000 HPLC system (Thermo Scientific, USA) with a Gemini® 5 µm C$_{18}$ 110 Å column (250 × 4.6 mm, Phenomenex, USA) and an amaZon SL ion trap mass spectrometer coupled with an electrospray ionization source (Bruker, USA). The medium-pressure liquid chromatography (MPLC) was performed using a dual-pump gradient system from Shanghai Lisui E-Tech Co., Ltd. (China) with ODS resin (50 µm, YMC, Japan). The semi-preparative HPLC was performed on a Dionex UltiMate 3000 HPLC system using a YMC-Pack ODS-A column (10.0 mm i.d. ×250 mm, 5 µm; YMC, Japan). The HRESIMS spectra were obtained with a Waters Synapt G2 TOF mass spectrometer (Waters Corporation, USA). 1D and 2D NMR spectra were recorded on Bruker AV 400/600 spectrometers (Bruker, USA) using the solvent signals (DMSO-$d_6$: $\delta_H$ 2.50/$\delta_C$ 39.5) as internal standards.

### Plant materials

*Lycium barbarum* was obtained from Zhongning County of Ningxia Hui Autonomous Region in China in August 2019. Different tissues were collected, immediately frozen in liquid nitrogen and preserved at −80 °C for further study. A voucher specimen (LYBA-NX-ZN) was deposited in the Institute of Traditional Chinese Medicine and Natural Products, College of Pharmacy, Jinan University, Guangzhou, China.

### Full-length isoform sequencing

Total RNA extraction and the library construction for full-length isoform sequencing (Iso-seq) were performed by Anoroad Co., Ltd. (Beijing, China). To obtain more reliable transcriptome data, mixed RNAs from three wolfberry tissues, including fruit, root and leaf were reverse transcribed into full-length cDNA by a SMARTer PCR cDNA Synthesis Kit (Clontech, CA, USA). KAPA HiFi PCR Kits were used to amply cDNA for size selection. Then SMRTbell library was then combined via a SMRTbell Template Prep kit (Pacific Biosciences, CA, USA). The PacBio Sequel system (Anoroad Co., Ltd.) was used to sequence the polymerase-bound template. Lastly, the high-quality isoforms underwent the standard annotation.

### Plasmids and strains

*E. coli* DH5α (Thermo Fisher Scientific, China) was used for cloning, while Rosetta (DE3) was used for expression. These strains were grown in Luria-Bertani (LB) medium supplemented with appropriate antibiotics. pGEX-2TK and pET-28b plasmids were purchased from the Sangon Biotech Co., Ltd. (Shanghai, China).

### Cloning, sequence alignment, and phylogenetic analysis of UGT genes

The total RNA of *L. barbarum* was extracted by Anoroad Co., Ltd. (Beijing, China) and reverse transcribed into cDNA by *TransScript*® One-Step gDNA Removal and cDNA Synthesis SuperMix (Transten, China) following the standard protocol. Genes were amplified by KOD One DNA polymerase (TOYOBO, Osaka, Japan) with the primers listed in Supplementary Table 10, and ligated into the pGEX-2TK vector *via* an In-Fusion® HD Cloning Kit.

The full-length protein sequences of all related enzymes were collected from GenBank and the JGI database (Supplementary Table 1), and aligned using ClustalW. A rooted maximum likelihood tree was generated using the Poisson model by the MEGA 6.0 software with bootstrapping for 1000 replicates.

## Expression and purification of recombinant protein

To expression of the recombinant protein for in vitro reaction and kinetic analysis, *E. coli* Rosetta (DE3) was transformed with the pGEX-2TK-LbUGT plasmids. Each transformant was incubated in LB medium supplemented with the corresponding antibiotic, at 37 °C/200 rpm for 16 h. Protein expression was induced by the addition of 0.4 mM Isopropyl $\beta$-D-thiogalactoside (IPTG) when the $OD_{600}$ reached 0.6, followed by further incubation with shaking at 160 rpm at 16 °C for 20 h. The cells were harvested by centrifugation (3 min, 8000 × $g$), and the LbUGT transformant was resuspended in 10 mM HEPES buffer (pH 7.0). After sonication on ice and centrifugation (30 min, 12,000 × $g$), the supernatant was purified by a GST-tag Protein Purification Kit (Beyotime, China) following the manufacturer's instruction. Protein purity was confirmed by SDS-PAGE, and a Bradford Protein Assay (Bio-Rad) was used to calculate protein concentration.

To expression and purification of LbUGTs for protein crystallization, the genes encoding LbUGT1 and LbUGT3 were cloned into pET-28b derivative vector pET-28b1 and pGEX-2TK derivative vector pGEX-2TK1, which both contain an 8×His tag and a TEV protease cleavage site at the N-terminus, respectively. These constructed plasmids were transformed into *E. coli* Rosetta (DE3) for overexpression, which was induced by the addition of 0.4 mM IPTG when the $OD_{600}$ reached 1.0, followed by incubation with shaking at 160 rpm at 16 °C for 20 h. Then the cells were harvested, resuspended, and lysed in buffer A (25 mM Tris, pH 8.0, 500 mM NaCl, and 5 mM β-Me). After centrifugation (30 min, 12,000 × $g$), the supernatant was purified by Ni-NTA chromatography, and the MBP-His-tag and GST-His-tag were removed by TEV. The target proteins were further purified by size-exclusive chromatography (Superdex 200 16/600, GE Healthcare) using buffer B (25 mM Tris, pH 8.0, 100 mM NaCl and 2 mM DTT). The proteins were concentrated to about 15 mg mL$^{-1}$ for protein crystallization.

## Enzyme activity measurements

The glycosyltransferase activity reaction mixture contained 2.5 mM UDP-Glc, 1.0 mM MgCl₂, 0.5 mM substrate, 50 µg of purified LbUGTs, and 10 mM HEPES (pH 7.0) in a final volume of 100 µL. The reactions were performed at 30 °C for 12 h and terminated by the addition of 200 µL of methanol. Subsequently, the reaction products were centrifuged at 12,000 × $g$ for 30 min to collect the supernatant, which was analyzed by LC-MS.

## Scale-up enzymatic reactions

The scaled-up enzymatic reactions were performed at 30 °C for 12 h in 10 mM HEPES buffer (pH 7.0), containing 2.5 mM UDP-Glc, 1.0 mM MgCl₂, 0.5 mM substrate, and 30 mL crude LbUGTs protein. The reactions were terminated by adding a 3-fold volume of methanol, and after centrifugation, the solvent was evaporated at reduced pressure and the residue was dissolved in methanol for purification.

## HPLC analysis and isolation of metabolites

Analytical HPLC was conducted with a linear gradient as follows: [MeOH (A) and H₂O containing 0.1% formic acid (B); 1 mL/min; 10%A (0 min) -10%A (10 min) -55%A (25 min) -100%A (26 min) -100%A (36 min).].

For the isolation of enzymatic products, scale-up enzymatic reactions were terminated by adding a three-fold volume of methanol to the enzyme-catalyzed system, and after centrifugation at 12,000 × $g$ for 30 min, the supernatant was collected and dried under reduced pressure. Then crude products were then subjected to MPLC by ODS column chromatography, eluted with a gradient of MeOH-H₂O, and further purified by preparative HPLC at 20–25% MeOH-H₂O (0.1% trifluoroacetic acid) to afford the main metabolites. The purified glycosylated products were dissolved in DMSO-$d_6$ and identified by 2D NMR spectroscopic analysis.

## Biochemical properties of LbUGT1-5

The biochemical properties of LbUGT1-5 were investigated using $N^1$-caffeoyl-$N^{10}$-dihydrocaffeoylspermidine (**1**) or $N^1$, $N^{10}$-bis-caffeoyl-spermidine (**2**) as the sugar acceptor and UDP-Glc as the sugar donor. To test the optimal reaction temperature for LbUGT1-5 activities, the catalytic assays were incubated at different temperatures (4 °C, 16 °C, 30 °C, 37 °C, 50 °C, and 70 °C) (Supplementary Fig. 6). To investigate the necessity of divalent metal ions for LbUGT1-5 activities, EDTA, MgCl₂, CaCl₂, MnCl₂, CoCl₂ and FeCl₂ were each used at a final concentration of 1 mM (Supplementary Fig. 7). Three parallel reactions were performed for each condition. The reaction was terminated by the addition of 100 µL of methanol and centrifuged at 12,000 × $g$ for 30 min to collect the supernatant for LC-MS analyses.

To determine the enzymatic kinetic parameters of recombinant LbUGT1-4 for compound **1** and LbUGT5 for compound **2**, the enzymatic assays were performed in 100 µL reaction mixtures containing 10 mM HEPES buffer (pH 7.0), 50 µg of purified recombinant LbUGTs, 2.5 mM UDP-Glc, 1.0 mM MgCl₂ and various concentrations of substrates (0.05, 0.1, 0.2, 0.5, 1, 2 and 3 mM). The enzymatic reactions were incubated at 30 °C for 1 h, which was selected based on the conversion rates, and quenched with 100 µL of methanol. After centrifugation at 12,000 × $g$ for 30 min, the supernatant was collected for LC-MS analyses. All reactions were performed in triplicate. The kinetic parameters $K_m$, $k_{cat}$, and $k_{cat}/K_m$ were determined with the Michaelis-Menten equation by Graphpad Prism 8 (Supplementary Fig. 8).

## Protein crystallization and structure determination

Crystal screening of LbUGT1 and LbUGT3 was performed at 19 °C by the sitting-drop vapor diffusion method. The proteins were mixed with 1 mM UDP or UDP-Glc and 10 mM substrate (**1**, **2** or **8**), and then incubated on ice for 1 hour to form complex. Crystals of LbUGT1 were obtained in a solution containing 0.2 M (NH₄)₂SO₄, 0.1 M HEPES (pH 7.5), 20% (w/v) PEG8000 and 10% (v/v) 2-propanol. Crystals of LbUGT3 were grown in a solution consisting of 0.1 M Bis Tris propane (pH 6.5), 0.2 M NaF, 20% PEG3350 and 4% (v/v) 2,5-Hexanediol. All crystals were flash-frozen in liquid nitrogen by using reservoir solution containing 10%-20% (v/v) glycerol as cryo-protectant. Data were collected on beamline BL19U1 of the Shanghai Synchrotron Radiation Facility (SSRF) and processed by xia2[56]. The structures of LbUGT1 and LbUGT3 were solved by molecular replacement using the search models predicted by AlphaFold2[57]. Iterative cycles of model rebuilding and refinement were achieved using PHENIX[58] and COOT[59]. The data collection and refinement statistics are presented in Supplementary Table 3.

## Molecular docking, molecular dynamics and site-directed mutagenesis

Molecular docking of LbUGT1-4 with **1**, **2**, **8** and UDP-Glc were investigated using Autodock 4.0[60]. The models were analyzed and screened according to the binding energies and conformations. All MD simulations were performed with AMBER22[61], using the AMBER ff14SB[62] force field for the protein and the TIP3P[63] model for solvent water molecules. The force field parameters of UDP-Glc and substrate **1**, **2**, or **8** were generated from the AMBER GAFF force field[64]. The partial atomic charges of UDP-Glc and substrate **1**, **2**, or **8** were obtained from the restrained electrostatic potential (RESP)[65] charge based on HF/6-31 G* calculations with the Gaussian 09 package[66]. The initial coordinates and topology files were generated by the

tleap program with neutralization and solvation. All models were treated with the same MD protocols by employing the periodic boundary conditions with cubic models. First, three steps of minimizations were carried out to relax the solvent and optimize the system. Each system was then heated from 0 to 300 K gradually under the NVT ensemble for 50 ps, and then another 100 ps of NPT ensemble MD simulations at 300 K and the target pressure of 1.0 atm were performed. In the NPT ensemble, the system temperature was controlled by the Langevin thermostat method. Afterward, a 50 ns MD simulation under the NVT ensemble was performed for each model. All hydrogen-containing bonds during the MD simulations were constrained using the SHAKE algorithm[67], and a time step of 2 fs was used for all simulations. A cutoff of 12 Å was set for both van der Waals and electrostatic interactions. Finally, all the trajectories were analyzed by CPPTRAJ program[68].

Mutagenesis of LbUGT1-5 were performed by PCR using pGEX-2TK-LbUGT1-5 as templates, with the primers listed in Supplementary Table 10. PCR products were purified, digested by DpnI, and transformed into *E. coli* DH5α. After confirmation by DNA sequencing, the mutant plasmids were transformed into *E. coli* Rosetta (DE3) for heterologous protein expression.

## Structural characterization of new compounds

Compound 7: Greenish oil; $[\alpha]^{25}_D$ − 22.3 (*c* 0.50, MeOH); UV (MeOH) $\lambda_{max}$ (log $\varepsilon$) 205 (1.67), 287 (1.26) nm; IR (KBr) $\nu_{max}$ 3369, 2945, 2873, 1685, 1518, 1439, 1281, 1203,1139, 1076, 805, 722 cm$^{-1}$; HRESIMS (positive) *m/z* 632.2804 [M + H]$^+$ (calcd for $C_{31}H_{42}N_3O_{11}$, 632.2819), see Supplementary Fig. 19A; $^1H$ and $^{13}C$ NMR data, see Supplementary Table 4.

Compound 9: Greenish oil; $[\alpha]^{25}_D$ − 29.1 (*c* 0.50, MeOH); UV (MeOH) $\lambda_{max}$ (log $\varepsilon$) 204 (1.69), 292 (1.63), 314 (1.70) nm; IR (KBr) $\nu_{max}$ 3397, 2946, 2879, 1654, 1517, 1439, 1283, 1203,1139, 1076, 827, 637 cm$^{-1}$; HRESIMS (positive) *m/z* 794.3350 [M + H]$^+$ (calcd for $C_{37}H_{52}N_3O_{16}$, 794.3348), see Supplementary Fig. 20A; $^1H$ and $^{13}C$ NMR data, see Supplementary Table 6.

Compound 15: Greenish oil; $[\alpha]^{25}_D$ − 16.8 (*c* 0.50, MeOH); UV (MeOH) $\lambda_{max}$ (log $\varepsilon$) 205 (1.68), 292 (1.50), 316 (1.53) nm; IR (KBr) $\nu_{max}$ 3306, 2933, 2879, 1603, 1512, 1439, 1382, 1263, 1070, 816 cm$^{-1}$; HRESIMS (positive) *m/z* 796.3510 [M + H]$^+$ (calcd for $C_{37}H_{54}N_3O_{16}$, 796.3504), see Supplementary Fig. 21A; $^1H$ and $^{13}C$ NMR data, see Supplementary Table 7.

Compound 16: Greenish oil; $[\alpha]^{25}_D$ − 25.5 (*c* 0.50, MeOH); UV (MeOH) $\lambda_{max}$ (log $\varepsilon$) 205 (1.87), 287 (1.34) nm; IR (KBr) $\nu_{max}$ 3417, 2955, 2882, 1682, 1433, 1388, 1203, 1139, 1170, 802, 722 cm$^{-1}$; HRESIMS (positive) *m/z* 794.3353 [M + H]$^+$ (calcd for $C_{37}H_{52}N_3O_{16}$, 794.3348), see Supplementary Fig. 22A; $^1H$ and $^{13}C$ NMR data, see Supplementary Table 8.

Compound 17: Greenish oil; $[\alpha]^{25}_D$ − 27.3 (*c* 0.50, MeOH); UV (MeOH) $\lambda_{max}$ (log $\varepsilon$) 205 (1.87), 288 (1.41) nm; IR (KBr) $\nu_{max}$ 3409, 2942, 2882, 1682, 1509, 1436, 1266, 1206, 1135, 805, 722 cm$^{-1}$; HRESIMS (positive) *m/z* 794.3347 [M + H]$^+$ (calcd for $C_{37}H_{52}N_3O_{16}$, 794.3348), see Supplementary Fig. 23A; $^1H$ and $^{13}C$ NMR data, see Supplementary Table 9.

## Reporting summary

Further information on research design is available in the Nature Portfolio Reporting Summary linked to this article.

## Data availability

All relevant data are available within the article and its Supplementary Information and from the corresponding authors on request. The structures of LbUGT1 and LbUGT3 have been deposited in the Protein Data Bank under codes 8WP5 and 8W53, respectively. The GenBank accession numbers for nucleotide sequences of LbUGT1, LbUGT2, LbUGT3, LbUGT4 and LbUGT5 are BankIt2754211 LbUGT1 OR684509, BankIt2754211 LbUGT2 OR684510, BankIt2754211 LbUGT3 OR684511, BankIt2754211 LbUGT4 OR684512 and BankIt2754211 LbUGT5 OR684513, respectively. The primers and predicted DNA sequences are given in Supplementary Tables 10 and 11. Source data are provided with this paper.

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

## Acknowledgements

This work was financially supported by grants from the National Key Research and Development Program of China (2023YFC3503900 to D.H.; 2018YFA0903200 to H.G.), the National Natural Science Foundation of China (81925037, 82321004, U22A20371 to H.G.; 32170060 to D.H.; 22177037 to J.M.L.; 22207039 to G.Q.W.), the Guangdong Natural Science Funds for Distinguished Young Scholar, China (2019B151502014 to D.H.; 2022B1515020028 to J.M.L.), the Guangdong International Science and Technology Cooperation Base, China (2021A0505020015 to H.G.), the Local Innovative and Research Teams Project of Guangdong Pearl River Talents Program, China (2017BT01Y036 to H.G.), the Innovative and Research Teams Project of Guangdong Higher Education Institution, China (2021KCXTD001 to H.G.), the Guangdong Basic and Applied Basic Research Foundation, China (2023A1515110388 to S.Y.L.), the Guangzhou Science and Technology Project, China (202206010020 to D.H.; SL2022A04J00367 to G.Q.W.) and Grant-in-Aid for Scientific Research from the Ministry of Education, Culture, Sports, Science and Technology, Japan (JSPS KAKENHI Grant Number JP20KK0173, JP21K18246, and JP23H00393 to I.A.). We thank the staffs at beamline BL19U1 of Shanghai Synchrotron Radiation Facility for help in data collection.

## Author contributions

H.G., I.A., and D.H. designed the research. S.-Y.L., G.-Q.W., L.L., and J.-L.G. performed the experiments. S.-Y.L., G.-Q.W., L.L., J.-L.G., Z.-Q.Z., Y.-H.W., J.-M.L., G.-D.C., D.H., I.A., and H.G. analyzed the data, and wrote the paper.

## Competing interests

The authors declare no competing interests.

## Additional information

[1]Institute of Traditional Chinese Medicine and Natural Products, College of Pharmacy/Guangdong Province Key Laboratory of Pharmacodynamic Constituents of Traditional Chinese Medicine and New Drugs Research/International Cooperative Laboratory of Traditional Chinese Medicine Modernization and Innovative Drug Development of Ministry of Education (MOE) of China, Jinan University, Guangzhou 510632, China. [2]Department of Radiology, The First Affiliated Hospital, Jinan University, Guangzhou 510632, China. [3]Graduate School of Pharmaceutical Sciences, the University of Tokyo, 7-3-1 Hongo, Bunkyo-ku, Tokyo 113-0033, Japan. [4]These authors contributed equally: Shao-Yang Li, Gao-Qian Wang, Liang Long. ✉e-mail: thudan@jnu.edu.cn; abei@mol.f.u-tokyo.ac.jp; tghao@jnu.edu.cn

