## [Peer Review File · Nature Communications]

Functional and structural dissection of glycosyltransferases underlying the glycodiversity of wolfberry-derived bioactive ingredients lycibarbarspermidinesREVIEWER COMMENTS

Reviewer #1 (Remarks to the Author):

The present work by Li et al. is a comprehensive investigation focused on characterizing the functions of glucosyltransferases involved in the biosynthesis of unusual phenolamide glycosides in wolfberry. The study involved a thorough elucidation of the structures of the extractable phenolamide glycosides, including metabolites that had not been described previously. Based on sequence identity comparisons with UDP-dependent glucosyltransferases (UGTs) across plant species, but with an emphasis on those that glucosylate phenylpropanoid acids and alcohols, five UDPGs from wolfberry were selected for further analysis. The corresponding genes were expressed heterologously in *E. coli*, the recombinant proteins purified and assayed under numerous conditions to further characterize the substrate and product specificities of these UGTs. These enzymological studies were complemented with the determination of crystal structures for two of these enzymes, UGT1 and UGT3. Since the structures were obtained without the phenolamide co-substrate, docking and molecular dynamics simulations were employed to explore the structural determinants of regioselectivity of these UGTs. Based on the analysis of enzyme variants that mirror UGT sequence variation, created at the gene level through site-directed mutagenesis, residues Y389 was identified as being of particular importance in controlling regiospecificity. The manuscript is well written and the narrative does a nice job of leading the reader through the different outcomes of the study. The figures are well-designed, informative, and readable. I write this very rarely, but I do not have any suggestions for improvements and believe that the work is certainly suited for publication in *Nature Communications*.

Reviewer #2 (Remarks to the Author):

The study by Shao-Yang Li et al identified five glucosyltransferases (LbUGT1-5) contributing to the glycodiversity of wolfberry-derived lycibarbarspermidines. These UGTs are described as the first phenolamide-type glucosyltransferases with different regioselectivity to dicaffeoylspermidines. LbUGT1 indicates very different site-selectivity to the caffeoyl (3-OH) and dihydrocaffeoyl (4-OH) groups of lycibararspermidines, and impressively, LbUGT3 can

catalyze a tandem sugar transfer to the ortho-dihydroxy groups on the caffeoyl moiety. Then, the structural basis raising the different regioselectivity of LbUGT1 and LbUGT3 were further explored by protein crystal structure analysis, molecular dynamic simulations, and site-directed mutagenesis. The manuscript is well prepared and informative, the results reported here provide insight into the enzymatic underpinnings of the structural diversity of lycibarbarspermidines in wolfberry, and this reviewer agree that the work is suitable for publication in Nature Communications.

Here just enclosed several concerns from this reviewer.

1. Do lycibarbarspermidines specifically occur in the fruits of wolfberry plant? How about the expression profiles of the five UGT genes reported here in different wolfberry tissues? Are these genes specifically in wolfberry fruits?
2. LbUGT2/4 also exhibit very specific and interesting regioselectivity, although the crystals of LbUGT2/4 were not obtained, is it possible to provide more information of the regioselectivity of LbUGT2/4 by protein modeling? Based on the results of LbUGT1 and LbUGT3, this seems possible.
3. In the Supplementary Figure 1B, the authors mentioned that the ortho-diglucosylated products (15, 16 and 17) produced by LbUGT3 are new compounds. In this case, are there any other lycibarbarspermidines with the ortho-diglucosyl moiety in wolfberry fruits? Is it possible to confirm the in vivo function of LbUGT3 by the expression of LbUGT3 gene in *Nicotiana benthamiana*?
4. Please check the configuration of the UDP-glucose moiety in Figures 5A and 6A.
5. Some of the enzymatic products were assigned by the comparison of retention time with standards. HPLC charts including the standards and MS spectra are recommended to be included in Supplementary Materials.
6. In Figures 6D and 6E, "Distane" should be "Distance"?

Reviewer #3 (Remarks to the Author):

This paper describes the characterization of a collection of glycosyltransferases from the Chinese medicinal plant, *Lycium barbarum* (wolfberry), that modify the terminal hydroxyl groups of dicaffeoylspermidines to form structurally diverse lycibarbarspermidines. The authors recently identified these enzymatic products as major constituents of wolfberry extracts with proposed and potential anti-oxidant and anti Alzheimers Disease activities. Known plant GT1 enzymes with activities for glycosylation of caffeic acid analogues were used to query gene sequences from wolfberry and 151 UGTs were identified. Additional filtering for similarity to other plant caffeic acid GTs led to a target list of 20 genes that were cloned, expressed and screened for activity against a representative caffeoyl-dihydrocaffeoylspermidine substrate. Five gene products showed activity and were further characterized for acceptor and donor specificity, kinetics, and crystal structures of two were determined in complex with UDP. Donor and acceptor complexes were modeled and mechanisms for the respective specificities were proposed.

Overall, this is a well-executed biochemical study on a relatively uncharacterized set of enzymes involved in natural product synthesis. The documentation of the choice of genes and enzymatic/structural/modeling characterization was well-done and mutagenesis and catalytic studies were consistent with the mechanistic models for specificity. There are no major concerns regarding the rigor of the biochemical studies.

The major concern regarding the manuscript is the significance of the studies on these enzymes involved in dicaffeoylspermidine glycosylation. The justification for choosing these enzymes was surprisingly superficial given the depth of studies performed. The lycibarbarspermidine products were proposed to be of biological significance as antioxidants and putative impact on AD in a fruit fly model. There is no biology described in the present paper and the rationale for the studies does not justify publication in a high impact journal. The present manuscript is of relatively narrow reader interest and is more appropriate for a far more specialized journal on natural product enzymology rather than the present journal.

Responsive letter to reviewers

REVIEWERS' COMMENTS:

Reviewer #1 (Remarks to the Author):

The present work by Li et al. is a comprehensive investigation focused on characterizing the functions of glucosyltransferases involved in the biosynthesis of unusual phenolamide glycosides in wolfberry. The study involved a thorough elucidation of the structures of the extractable phenolamide glycosides, including metabolites that had not been described previously. Based on sequence identity comparisons with UDP-dependent glucosyltransferases (UGTs) across plant species, but with an emphasis on those that glucosylate phenylpropanoid acids and alcohols, five UDPGs from wolfberry were selected for further analysis. The corresponding genes were expressed heterologously in *E. coli*, the recombinant proteins purified and assayed under numerous conditions to further characterize the substrate and product specificities of these UGTs. These enzymological studies were complemented with the determination of crystal structures for two of these enzymes, UGT1 and UGT3. Since the structures were obtained without the phenolamide co-substrate, docking and molecular dynamics simulations were employed to explore the structural determinants of regioselectivity of these UGTs. Based on the analysis of enzyme variants that mirror UGT sequence variation, created at the gene level through site-directed mutagenesis, residues Y389 was identified as being of particular importance in controlling regiospecificity. The manuscript is well written and the narrative does a nice job of leading the reader through the different outcomes of the study. The figures are well-designed, informative, and readable. I write this very rarely, but I do not have any suggestions for improvements and believe that the work is certainly suited for publication in *Nature Communications*.

Response:

We are grateful to the reviewer for taking the time to provide us with the positive feedback and valuable recommendations regarding our research.

Reviewer #2 (Remarks to the Author):

The study by Shao-Yang Li et al identified five glycosyltransferases (LbUGT1-5) contributing to the glycodiversity of wolfberry-derived lycibarbarspermidines. These UGTs are described as the first phenolamide-type glycosyltransferases with different regioselectivity to dicaffeoylspermidines. LbUGT1 indicates very different site-selectivity to the caffeoyl (3-OH) and dihydrocaffeoyl (4-OH) groups of lycibararspermidines, and impressively, LbUGT3 can catalyze a tandem sugar transfer to the ortho-dihydroxy groups on the caffeoyl moiety. Then, the structural basis raising the different regioselectivity of LbUGT1 and LbUGT3 were further explored by protein crystal structure analysis, molecular dynamic simulations, and site-directed mutagenesis. The manuscript is well prepared and informative, the results reported here provide insight into the enzymatic underpinnings of the structural diversity of lycibarbarspermidines in wolfberry, and this reviewer agree that the work is suitable for publication in Nature Communications.

Response:

We greatly appreciate the reviewer for carefully reading our manuscript, and providing valuable suggestions and comments to improve our manuscript.

Here just enclosed several concerns from this reviewer.

1. Do lycibarbarspermidines specifically occur in the fruits of wolfberry plant? How about the expression profiles of the five UGT genes reported here in different wolfberry tissues? Are these genes specifically in wolfberry fruits?

Response:

(1) The dicaffeoylspermidine glycosides lycibarbarspermidines are exclusively found in the fruits of wolfberry plants through metabolome analysis of various tissues including fruits, leaves, and root bark. As shown in Fig. R1A, wolfberry fruits contain phenylpropionyl-spermine (glycoside), lycibarbarspermidine, phenylpropionyl-spermidine (aglycone) and flavonoids; wolfberry leaves contain phenylpropionyl-agmatine/putrescine (aglycone), phenylpropionyl-spermine (aglycone), chlorogenic acids, phenylpropionyl-spermidine (aglycone) and flavonoids; while the root bark contains phenylpropionyl-agmatine/putrescine (aglycone) and phenylpropionyl-spermine (aglycone). The chemical structures of most chromatographic peaks from wolfberry fruits,

leaf and root bark were further identified by comparison with authentic compounds (Fig. R1B-R1E).

These results clearly indicate that lycibarbarspermidines are solely present in the fruit.

Fig. R1 Metabolome analysis of the various tissues of wolfberry. (A) Metabolome analysis of wolfberry fruit, leaf and root bark. (B-E) Identification of metabolites from different tissues of wolfberry.

(2) In our previously analyzed transcriptomes, we observed the expression of LbUGT1-5 in various tissues of wolfberry, with a higher level of expression for LbUGT1-3 and 5 in the fruit (Fig. R2). Particularly, LbUGT1-3 demonstrated relatively high catalytic activity among the five enzymes (Table 1), which likely contributes to the predominant accumulation of lycibarbarspermidines in wolfberry fruit. Despite observing the expression of LbUGT1-5 in leaves and roots, we did not detect the presence of lycibarbarspermidines in these tissues. This discrepancy may be attributed to a specialized transport mechanism in wolfberry plants that facilitates the storage of glycosylated phenylpropionyl-spermidine specifically in the fruit.

Fig. R2 Transcriptome expression patterns of LbUGT1-5 in different tissues of wolfberry plant.

2. LbUGT2/4 also exhibit very specific and interesting regioselectivity, although the crystals of LbUGT2/4 were not obtained, is it possible to provide more information of the regioselectivity of LbUGT2/4 by protein modeling? Based on the results of LbUGT1 and LbUGT3, this seems possible.

Response:

Thank you for the suggestions. As we were unable to obtain crystals of LbUGT2/4, we used AlphaFold2 to model the structures of LbUGT2 and LbUGT4. Following the methods used for LbUGT1 and LbUGT3, UDP-Glc and substrate **2** were docked into the protein structures, followed by MD simulations to acquire the stable binding conformations. Our findings show that in both LbUGT2 and LbUGT4 structures, the 4'-OH group of **2** forms a stable hydrogen bond with the catalytic histidine (H16 and H21) and is in close proximity to the acetal C atom of UDP-Glc, consistent with their glycosylation activity towards the 4-OH of the caffeoyl group. Notably, F382 in LbUGT2 and Y388 in LbUGT4, which correspond to the Y389 in LbUGT1 and Y390 in LbUGT3, engage in hydrophobic interaction and hydrogen bond with **2** (Supplementary Fig. 15). Subsequent substitution of the conserved tyrosine residue with A led to drastically decreased activity or even complete inactivation, indicating a crucial role for this conserved tyrosine residue in the function of LbUGT enzymes (Fig. 6F). These results have been integrated into the manuscript (page 11, lines 290-303), and corresponding data are included in Supplementary Fig. 15.

Supplementary Figure 15. The binding conformations of LbUGT2 (A) and LbUGT4 (B). The common substrate **2** shown in cyan and UDP-Glc shown in yellow. The key amino acids were labeled by green sticks and the hydrogen bonds were shown with the purple dash.

3. In the Supplementary Figure 1B, the authors mentioned that the ortho-diglucosylated products (**15**, **16** and **17**) produced by LbUGT3 are new compounds. In this case, are there any other lycibarbarspermidines with the ortho-diglucosyl moiety in wolfberry fruits? Is it possible to confirm the *in vivo* function of LbUGT3 by the expression of LbUGT3 gene in *Nicotiana benthamiana*?

Response:

(1) We greatly appreciate your valuable suggestion. We conducted an investigation into the presence and diversity of lycibarbarspermidines with the ortho-diglucosyl moiety in wolfberry fruit utilizing high-resolution mass spectrometry (HR-MS). Specifically, the $[M+H]^+$ ion of compound **15** was fragmented into ions m/z 544.2024, 382.1494, and 220.0967, corresponding to the unit of N^1 -diglucosyl caffeoyl, N^1 -monoglucosyl caffeoyl, and N^1 -caffeoyl, respectively. The ion at m/z 544.2024 was particularly significant, serving as a potential diagnostic product ion of ortho-diglucosyl caffeoyl unit (Fig. R3A and B). Utilizing this diagnostic ion, we screened out multiple lycibarbarspermidine diglucosides (m/z : 796.3489/794.3344) and triglucosides (m/z : 958.4020/956.3865) within the lycibarbarspermidine-enriched fractions of wolfberry at the retention time ranged from 32.59 to 36.92 min (Fig. R3C). The representative MS/MS spectra of a triglucoside is shown in Fig. R3D, strongly suggesting the presence of lycibarbarspermidines with the ortho-diglucosylation moiety as compound **15**. However, the current approach cannot rule out the possibility that the sugar group may bind to the glucose directly, isolation of these compounds

might be necessary for further confirmation.

Fig. R3 Proposed fragmentation pathway of ortho-diglucosylated product **15** (A) and its MS/MS spectra (B). (C) The metabolite analysis of lycibarbarspermidine-enriched fraction of wolfberry fruits, in which the region marked by the red box contains multiple lycibarbarspermidine diglucosides and triglucosides containing m/z 544.2024 ions. (D) The representative MS/MS spectra of a lycibarbarspermidine triglucoside.

(2) For the *in vivo* functional analysis of LbUGT3, we generated the recombinant plasmid pY622-LbUGT3 and control plasmid pY622-GFP. Subsequently, these plasmids were individually introduced into *Agrobacterium tumefaciens* GV3101, and the resulting *Agrobacterium* suspension was transiently transfected into 5-week-old *N. benthamiana* leaves. After a three-day incubation period, successful expression of GFP fluorescent protein was confirmed through UV light irradiation. Following this confirmation, substrate **1** was injected into the *Agrobacterium*-infiltrated leaves containing both the LbUGT3 group and GFP group. (Fig. R4A). Leaves were crushed after three days by grinding with liquid nitrogen and extracted with methanol for HR-MS analysis. Compared with the GFP group, *N. benthamiana* leaves expressing LbUGT3 exhibited the conversion of substrate **1** to diglucoside **15**, providing compelling evidence for the *in vivo* tandem glycosylation activity of LbUGT3 on the ortho-dihydroxyl group of the caffeoyl moiety (Fig. R4B-D).

Fig. R4 (A) The *N. benthamiana* leaf exhibited the GFP fluorescent under 410 nm UV light. (B) EIC for ortho-diglucosylated product **15** (m/z : 796.3495 \pm 0.0010) produced by transiently expressed LbUGT3 in *N. benthamiana* leaves, compared to the authentic standard of compound **15**. (C) Reactions catalyzed by LbUGT3. (D) MS/MS fragmentation of standard **15** compared to the fragmentation of the compound **15** produced by transiently expressed LbUGT3 in *N. benthamiana* leaves.

Method

Functional Verification of LbUGT3 in *N. benthamiana*: The *N. benthamiana* plants were grown in soil containing peat soil and vermiculite with a 16 h light cycle at room temperature. Plants were grown for 5 weeks before *Agrobacterium* infiltration.

The control plasmid pY66-GFP was digested with *Pst*I and *Sac*I and assembled with the PCR product of LbUGT3. The assembly mixtures were transformed into competent *E. coli* DH5 α . Confirmed recombinant plasmids were transformed into *A. tumefaciens* GV3101 using the freeze-thaw method (Sangon Biotech, China). The GV3101 (pY66-LbUGT3/GFP) cells were suspended in MES buffer and it was mixed with an equal concentration of GV3101 (pSoup-p19) cells to obtain *Agrobacterium* suspensions (OD₆₀₀ = 0.2) (the plasmids pY66-GFP and pSoup-p19 were kindly provided by Changqing Yang, Chinese Academy of Agricultural Sciences). The *Agrobacterium* suspensions were incubated at 24 °C for 3 h and then infiltrated into *N. benthamiana* leaves using a

1 mL syringe. After three days, the successful expression of GFP fluorescent protein was verified by irradiation of tobacco leaves with a 410 nm UV lamp, followed by infiltration of previously *Agrobacterium*-infiltrated leaves with 500 μ M substrate **1**. Leaves were crushed after three days by grinding with liquid nitrogen and extracted with methanol for HR-MS analysis.

4. Please check the configuration of the UDP-glucose moiety in Figures 5A and 6A.

Response:

Thanks for the reviewer's notice. We have modified the configuration of the UDP-glucose moiety (Figures 5A and 6A).

5. Some of the enzymatic products were assigned by the comparison of retention time with standards. HPLC charts including the standards and MS spectra are recommended to be included in Supplementary Materials.

Response:

We have supplemented the HPLC charts including the standards and MS spectra (Supplementary Fig. 4).

6. In Figures 6D and 6E, "Distane" should be "Distance"?

Response:

We are sorry for this mistake. We have corrected the "Distane" to "Distance" (Figures 6D and 6E).

Reviewer #3 (Remarks to the Author):

This paper describes the characterization of a collection of glycosyltransferases from the Chinese medicinal plant, *Lycium barbarum* (wolfberry), that modify the terminal hydroxyl groups of dicaffeoylspermidines to form structurally diverse lycibarbarspermidines. The authors recently identified these enzymatic products as major constituents of wolfberry extracts with proposed and potential anti-oxidant and anti Alzheimers Disease activities. Known plant GT1 enzymes with activities for glycosylation of caffeic acid analogues were used to query gene sequences from

wolfberry and 151 UGTs were identified. Additional filtering for similarity to other plant caffeic acid GTs led to a target list of 20 genes that were cloned, expressed and screened for activity against a representative caffeoyl-dihydrocaffeoylspermidine substrate. Five gene products showed activity and were further characterized for acceptor and donor specificity, kinetics, and crystal structures of two were determined in complex with UDP. Donor and acceptor complexes were modeled and mechanisms for the respective specificities were proposed.

Overall, this is a well-executed biochemical study on a relatively uncharacterized set of enzymes involved in natural product synthesis. The documentation of the choice of genes and enzymatic/structural/modeling characterization was well-done and mutagenesis and catalytic studies were consistent with the mechanistic models for specificity. There are no major concerns regarding the rigor of the biochemical studies.

The major concern regarding the manuscript is the significance of the studies on these enzymes involved in dicaffeoylspermidine glycosylation. The justification for choosing these enzymes was surprisingly superficial given the depth of studies performed. The lycibarbarspermidine products were proposed to be of biological significance as antioxidants and putative impact on AD in a fruit fly model. There is no biology described in the present paper and the rationale for the studies does not justify publication in a high impact journal. The present manuscript is of relatively narrow reader interest and is more appropriate for a far more specialized journal on natural product enzymology rather than the present journal.

Response:

We sincerely appreciate the reviewer's favorable feedback on our biochemical research and their high assessment of the thoroughness of our current studies.

We would like to address your concerns regarding the significance of lycibarbarspermidines. Since our initial discovery of lycibarbarspermidines as a major class of bioactive ingredients in wolfberry in 2016 (*Journal of Agricultural and Food Chemistry*, 2016, 64, 2223-223; *World Journal of Traditional Chinese Medicine*, 2016, 2, 1-5), our research group has been dedicated to investigating their application in various diseases. We have conducted a series of pharmacological studies focusing on neurodegenerative disorders including Alzheimer's disease and Parkinson's disease, as well as alcoholic liver disease, utilizing animal models including rats and mice. Our findings have demonstrated that lycibarbarspermidines exhibit significant therapeutic effects against

these diseases. Additionally, we have identified their protein targets using ABPP approaches, and these results are currently being prepared for publication. The potential of lycibarbarspermidines as an innovative drug has led us to apply for a series of PCT patents in Europe, USA, Japan, and China (EP3406591A1, EP3406620B1, US10457702B2, US11208427B2, JP6902757, JP6923861, ZL201610033609.8 and ZL201610032769.0), which have been granted. Therefore, biosynthetic characterization of lycibarbarspermidines hold great significance and are expected to exert substantial influence in the field.

In addition, lycibarbarspermidines are distinctive derivatives of spermidine, a widely distributed natural polyamine present in all living organisms from bacteria to humans. Spermidine plays an essential role in cell growth, proliferation, and tissue regeneration. As a natural inducer of autophagy and an anti-aging compound, spermidine has been extensively reviewed in *Science* (2018, 359, 410), with 556 citations to date and recognition as a highly cited paper, indicating significant interest in spermidine research. The exploration of lycibarbarspermidines as part of the spermidine family can further advance our understanding and utilization of these compounds.

Moreover, lycibarbarspermidines are the primary bioactive compounds found in wolfberry. Wolfberry has gained worldwide popularity as a natural health food and is widely recognized as a superfood. In Europe, the United States, and China, wolfberry can be consumed directly or processed into various forms such as wolfberry tea, wine, juice, seed oil, and cream to meet diverse consumer demands. Only in China, the cultivation area of wolfberry reached 286,000 hectares with a yield of over 400,000 tons and a total production value of 36 billion yuan. Our study sheds light on the molecular basis for lycibarbarspermidines diversity in wolfberry, which will serve as a foundation for breeding and cultivation as well as the development of innovative wolfberry products.

REVIEWERS' COMMENTS

Reviewer #2 (Remarks to the Author):

The authors have addressed all the concerns raised by this reviewer, and I am delighted to agree with the acceptance of this manuscript by Nature Communications.